# Characteristics and Expression Analysis of Invertase Gene Family in Common Wheat (*Triticum aestivum* L.)

**DOI:** 10.3390/genes14010041

**Published:** 2022-12-23

**Authors:** Chao Wang, Guanghao Wang, Xinyu Wen, Xiaojian Qu, Yaoyuan Zhang, Xiangyu Zhang, Pingchuan Deng, Chunhuan Chen, Wanquan Ji, Hong Zhang

**Affiliations:** 1State Key Laboratory of Crop Stress Biology for Arid Areas, College of Agronomy, Northwest A&F University, Xianyang 712100, China; 2Shaanxi Research Station of Crop Gene Resources and Germplasm Enhancement, Ministry of Agriculture, Xianyang 712100, China

**Keywords:** wheat, genome-wide, invertase, expression profiles

## Abstract

Invertase (INV) irreversibly catalyzes the conversion of sucrose into glucose and fructose, playing important role in plant development and stress tolerance. However, the functions of INV genes in wheat have been less studied. In this study, a total of 126 TaINV genes were identified using a genome-wide search method, which could be classified into five classes (TaCWI-α, TaCWI-β, TaCI-α, TaCI-β, and TaVI) based on phylogenetic relationship. A total of 101 *TaINVs* were collinear with their ancestors in the synteny analysis, and we speculated that polyploidy events were the main force in the expansion of the *TaINV* gene family. Compared with TaCI, TaCWI and TaVI are more similar in gene structure and protein properties. Transcriptome sequencing analysis showed that *TaINVs* expressed in multiple tissues with different expression levels. Among 19 tissue-specific expressed *TaINVs*, 12 *TaINVs* showed grain-specific expression pattern and might play an important role in wheat grain development. In addition, qRT-PCR results further confirmed that *TaCWI50* and *TaVI27* show different expression in grain weight NILs. Our results demonstrated that the high expression of *TaCWI50* and *TaVI27* may be associated with a larger TGW phenotype. This work provides the foundations for understanding the grain development mechanism.

## 1. Introduction

Sucrose is synthesized in the source leaves, loaded into the phloem for long-distance translocation, and unloaded into non-photosynthetic sink tissues such as roots, stems, flowers, fruits, and seeds [1]. Many studies have shown that sucrose degradation contributes to plant growth and development such as seed germination, root elongation, pollen development, fruit set, and ultimately seed size [2,3]. Two types of enzymes are responsible for the degradation of sucrose: sucrose synthase (Sus; EC 2.4.1.13) and invertase (INV; EC 3.2.1.26). Sus reversibly catalyzes the hydrolysis of sucrose into UDP-glucose and fructose and plays a crucial role in cell wall and cellulose biosynthesis [4]. In contrast, invertase irreversibly degrades sucrose into glucose and fructose and plays an essential role in carbon partitioning, sink development, sugar signaling, and the stress response. 

The invertase genes have been identified by genome-wide analyses in many plant species, including *Arabidopsis thaliana* [5], *Oryza sativa* [6], *Zea may* [7], *Glycine max* [8], *Saccharum officinarum* [9], *Pisum sativum* [10], *Solanum tuberosum* [11], *Malus domestica* [12], and *Populus* [13]. The invertase isoenzyme forms can be classified into two subfamilies, neutral/alkaline invertases (A/N-INVs) and acid invertases (Ac-INVs), based on their optimum pH [3]. Ac-INVs are glycosylated proteins classified in glycoside hydrolase family 32 (GH32). Ac-Invs contains three conserved motifs of NDPNG (β-fructofuranosidases motif), RDP, and WECP(V)D, which are necessary for catalytic activity [6]. Moreover, three amino acids (DPN) of the β-fructofuranosidases motif were coded by the smallest exon known to plant species and skipped in cold stressed potato [14]. According to their localization in apoplast and vacuole, Ac-INVs can be further divided into cell wall invertase (CWI) and vacuolar invertase (VI) [15]. Meanwhile, A/N-INVs are nonglycosylated proteins classified in glycoside hydrolase family 100 (GH100) and specifically hydrolyze sucrose. However, A/N-Invs are better known as cytoplasm invertases (CI), due to their localization in cytoplasm [16]. The CIs can be further divided into CI-α and CI-β subgroups.

The invertase genes perform various roles in plants [6,7,8,17]. Numerous studies have shown that CIs are essential for normal root development [18,19,20]. Mutation of *CYT-INV1* in arabidopsis and rice resulted in a shorter primary root [19,21]. Similar to CIs, the role of VI in root development has also been found. In detail, allelic differences in *AtVI2* cause differences in root length between divergent genotypes [22]. As for grain development and yield formation, mutation of the CWI gene *INCW2* in maize reduced maize grain weight by 70% [23], and mutation of *OsINV3* lead to a shorter panicle with lighter and smaller grains [24]. Conversely, overexpression of CWI enhanced grain filling and yield in rice and maize [25,26]. INVs are also play important roles in abiotic responses. *CsVI2* was upregulated by drought and the overexpression of *CsVI2* could enhanced drought tolerance in cucumber seedlings [27]. Recently, new function regarding CIs have been discovered. In detail, *At-A/N-InvC* plays an important role in the maintenance of mitochondrial reactive oxygen species homeostasis and the regulation of germination and flowering [28]. *At-A/N-InvE* mainly maintains the carbon balance between plastids and cytoplasm [29]. *AtVI2* may play an important role in regulating stomatal opening in Arabidopsis [30]. Knockdown of vacuolar invertase gene expression lowers the cold-induced sweetening in potatoes [31]. The gene expression and enzyme activity of INV are influenced by a variety of factors, like sugars, plant growth regulators (GA3, IAA, ABA, and CTK), abiotic stress, pathogen infection, and endogenous inhibitors [32].

Common wheat (*Triticum aestivum* L.) (2n = 6x = 42) is one of the most important planted crops worldwide. As the third largest cereal crop, common wheat accounts for approximately 20% of the total dietary calories and proteins worldwide [33]. With the continuous growth of the population, the demand for wheat is increasing, so improving the yield has always been one of the main tasks of wheat breeding [34]. The new yield related gene mined is helpful to improve wheat yield through gene editing technology [35]; genome-wide gene family analysis laid the foundation to excavate key genes [36]. In wheat, *TaCwi-A1*, located on wheat chromosome 2A, could explain 4.8% of phenotypic variance for kernel weight [37]. In addition, two CWI genes (*TaCWI-4A* and *TaCWI-5D*) were confirmed to be associated with TGW (thousand grain weight), and compared with Hap-4A-C haplotype, Hap-4A-T haplotype was associated with larger TGW and fewer seeds [38]. A recent study showed that the *Ta-A/N-Inv1* gene negatively regulates the disease resistance to wheat stripe rust [39]. This means that INV played important functional roles in wheat development and adaptation to stress.

The release of the high-quality genome sequence of hexaploid wheat cv. Chinese Spring (CS) has facilitated the identification of wheat gene families at the genome levels [40]. Although Webster et al. identified eight CWI genes and deduced their functions in response to water deficit [41], the identification of the CWI family was incomplete in term of its complexity. Herein, a total of 126 INV genes were identified in wheat using the whole-genome identification method. Moreover, a series of bioinformatic approaches were used to unravel the gene structure features, chromosomal locations, phylogenetic relationships, synteny, and expression patterns to highlight the potential functional diversity. This will lay a foundation for the functional analysis of the *TaINV* gene in the future.

## 2. Materials and Methods

### 2.1. Identification the Invertase Gene Family in Wheat and its Ancestors

The genomic sequences, protein sequences, and annotation files of *T. aestivum* (IWGSC RefSeq v2.1) were obtained from the IWGSC dataset (https://wheat-urgi.versailles.inra.fr/Seq-Repository/Assemblies, accessed on 16 September 2020). The genomic sequences, protein sequences, and annotation files of *Triticum urartu* (G1812), *Triticum dicoccoides* (WEWSeq_v.1.0), and *Aegilops tauschii* (v4.0) were downloaded from the Ensembl Plants Database (http://plants.ensembl.org/index.html, accessed on 19 August 2022)). The longest splice variant of these four species was selected as a representative for subsequent analysis. The invertase protein sequence of rice, maize, Aarabidopsis, and soybean were download from Ensemble Plants.

To identify INV members in common wheat and its relatives, two methods were used, namely, HMMER search and blastP analysis. In detail, the hidden Markov model (HMM) seed files including Glyco_hydro_32N (PF0000251), Glyco_hydro_32C (PF08244), and Glyco_hydro_100 (PF12899) were downloaded from the Pfam website (https://pfam-legacy.xfam.org/, accessed on 25 November 2021) and were used for HMMER search with a threshold of e < 1 × e^−10^. Meanwhile, the protein sequences of 17 INVs in Arabidopsis and 18 INVs in rice were used as queries to search against the local protein database using e < 1 × e^−5^, querry coverage >70% and identity >50% as the threshold. Finally, all candidate invertase obtained from the above methods were identified by CD-search (https://www.ncbi.nlm.nih.gov/Structure/cdd/wrpsb.cgi, accessed on 25 November 2021).

### 2.2. Phylogenetic Trees Construction and Collinearity Analysis

To analyze evolutionary relationships of the INV gene family, a phylogenetic tree was constructed based on the alignment of the 346 putative full-length amino acid sequences of rice, maize, soybean, Arabidopsis, common wheat, and its ancestors (*T. urartu*, *T. dicoccoides*, and *Ae. tauschii*). The phylogenetic tree was constructed with MEGA X software using the Maximum-like method, with default parameters and the bootstrap test executed by 1000 replications [42,43,44]. Then, the phylogenetic tree was optimized using the Interactive Tree of Life (iTOL, v6) (https://itol.embl.de/upload.cgi, accessed on 1 November 2022). At the same time, a phylogenetic tree of TaINV was also construct with the same method.

Collinearity analysis between wheat, rice, maize, Arabidopsis, soybean, and its relatives comprising (*T. urartu*, *T. dicoccoides*, and *Ae. tauschii*) were analyzed using the MCScanX (Multiple Collinearity Scan toolkit: http://chibba.pgml.uga.edu/mcscan2/) with a threshold e < 1 × e^−5^ and 5 hits. A collinearity map was drawn by Dual Systeny Plotter software in TBtools [45].

### 2.3. Chromosomal Location, and Gene Duplication Analysis of TaINVs

The physical location information of all *TaINV* genes was obtained from gff3 files and displayed using TBtools. Gene replication, including tandem replication and fragment replication, was obtained by the results of collinearity analysis described above. To understand the evolutionary constraints on *TaINV* genes, the substitution rate of nonsynonymous (Ka) and synonymous (Ks) was calculated. The timing of duplication events formula T = Ks/2λ × 10^−6^ Mya was used to calculate divergence time (T) in millions of years (Mya), where λ = 6.5 × 10^−9^ [46].

### 2.4. Protein Characterization and Subcellular Position Prediction

The molecular weight (MW), isoelectric point (pI), instability index (II), and grand average hydropathicity (GRAVY) were analyzed using the ExPASy online tools (https://web.expasy.org/cgi-bin/compute_pi/pi_tool, accessed on 14 December 2021) [47]. At the same time, conserved motifs were analyzed using the MEME (Multiple Expectation Maximization for Motif Elicitation) online program (http://meme-suite.org/tools/meme, version 5.5.0, accessed on 5 August 2022). The repeat number was set to 0 or 1, the maximum number of motifs to 10, the motif width was 6 to 50, and the rest of the run parameters to system default.

The subcellular location of the TaINV proteins were predicted with the ProtComp 9.0 web tools (http://www.softberry.com/, accessed on 14 December 2021). To further understand the function of N-terminal sequence, SignalP 6.0 (https://services.healthtech.dtu.dk/service.php?SignalP, accessed on 14 December 2021) was mainly used to predict the existence of signal peptide and TargetP-2.0 (https://services.healthtech.dtu.dk/service.php?TargetP, accessed on 14 December 2021) was used to predict the existence of mitochondrial transport peptide (mTP), chloroplast transport peptide (cTP), or cystoid coelomotor transport peptide (lTP). The presence of transmembrane helices (TMH) in proteins was predicted using TMHMM (http://www.cbs.dtu.dk/services/TMHMM/, accessed on 14 December 2021).

### 2.5. Gene Structure and Protomter Regions Analysis

The exon/intron information of all *TaINV* genes were obtained from gff3 files and displayed using TBtools. The 2 kb sequence upstream of the start codon is considered the promoter region and is extracted from the wheat genome (version 2.1). The cis-elements in the promoter sequences were identified in the PlantCARE database with the default parameters (https://bioinformatics.psb.ugent.be/webtools/plantcare/html/, accessed on 14 December 2021) [48].

### 2.6. Expression Patterns Analysis of TaINVs

To understand the potential functions of *TaINVs*, multiple transcriptome data involved in different tissues (root, leaf, stem, spike, and grain) and of wheat cultivar Chinese Spring and responses to common stress (drought, heat, low temperature, *Fusarium graminearum* (*Fhb*), *Blumeria graminis f. sp. Tritici* (*Bgt*) and *Puccinia striiformis f. sp. Tritici* (*Pst*)) were downloaded from the Wheat Expression Browser (http://www.wheat-expression.com/, accessed on 14 December 2021) [49]. The spatio-temporal expression of *TaINV* genes were visualized by TBtools based on log_2_ (TPM + 1) values. For convenience, the heat map of genes in response to stress was drawn based on the log_2_ (TPM + 1) ratios of treatment to control groups.

### 2.7. RNA Extraction and Quantitative Real-Time PCR

A pair of near isogenic lines (NILs), RHL81-L (TGW: 52.58 g) and RHL81-S (TGW: 33.35 g), produced by selfing of the RHL81, were used in expression analysis. Grains were marked at pollen mergence stage. Each sample contained 10 plants of uniform phenotype and the grains in the middle of spikelets shall be sampled at 4, 7, and 10 DPA (day post anthesis) and stored at −80 °C for RNA extraction. Total RNA was extracted using TRIZOL reagent and cDNA synthesis was performed with a reverse transcription reagent kit according to the manufacturers’ protocols.

Nine *TaINV* genes were analyzed by a real-time quantitative PCR (qRT-PCR). Specific primers of *TaINV* and *TaActin* gene (i.e.; internal reference) were designed with the Primer 5 program and shown in Appendix A. qRT-PCR was performed on the QuantStudio 7 Flex Real-Time PCR System (Life Technologies Corporation, Foster, CA, USA) with Quantitect SYBR I Green (TaKaRa Biotechnology Corporation, Dalian, China) according to the manufacturer’s protocol. Three independent biological replicates were prepared for each time points. The RHL81-S sample at 4 DPA was used as a control in the calculation of relative expression levels using the 2^−ΔΔCT^ method. All data were analyzed and evaluated by GraphPad Prism (version 5, San Diego, CA, USA). Two-tailed Student’s *t* test was performed to compare differences in gene expression.

### 2.8. Subcellular Localization of TaCWI50

The full-length coding sequence (CDS) of *TaCWI50* was amplified with the specific primers (TaCWI50-GFP-F/R) and inserted into *pYJGFP* vector using the ClonExpress II One Step Cloning Kit (Vazyme, Nanjing, China). The 35S::*TaCWI50*-GFP recombinant plasmid was used for substant experiments after sequenced. The recombinant plasmid 35S::*TaCWI50*-GFP and the control vector 35S::GFP were transformed into *Nicotiana benthamiana* leaf epidermal cells using *Agrobacterium tumefaciens* (GV3101) according to a previously described method [50]. The transformed epidermal cells were lifted, and the GFP signals were evaluated under an Olympus fuorescence microscope after 48 h incubation.

## 3. Results

### 3.1. Identification of INV Gene Family in Common Wheat and Its Ancestors

For convenience, Ta, Tu, Td, and Aet were used as prefixes before the names of INV genes from *T. aestivum*, *T. urartu*, *T. dicoccoides*, and *Ae. tauschii*, respectively. To identify the INV genes in common wheat and its ancestors, four HMM seed files were obtained: Glyco_hydro_32C (PF08244), Glyco_hydro_32N (PF00251), Domain of unknown function (PF11867), and Glyco_hydro_100 (PF12899) by entering the INV protein sequence of Arabidopsis and rice. Then, 375 protein sequences were obtained by HMMsearch and BlastP. After removing the sequences with incomplete domain determined by Pfam and CDD databases, 126 TaINVs, 35 TuINVs, 65 TdINVs, and 32 AetINVs were identified and used for subsequent analysis (Appendix A).

### 3.2. Phylogenetic Analysis of the Invertase Family

A phylogenetic tree was constructed based on the alignment of the 346 putative full-length amino acid sequences of rice, maize, soybean, Arabidopsis, common wheat, and its ancestors (*T. urartu*, *T. dicoccoides*, and *Ae. tauschii*) (Appendix A). The phylogenetic tree was classified into five major groups with high confidence: CI-α, CI-β, CWI-α, CWI-β, and VI (Figure 1). The five groups contain 40, 47, 64, 90, and 105 members, respectively. According to the position of genes on the evolutionary tree, 126 TaINVs were named TaCWI1~61, TaVI1~44 and TaCI1~21; 35 TuINVs were named TuCWI1~15, TuVI1~14 and TuCI1~6; 65 TdINVs were named TdCWI1~29, TdVI1~22 and TdCI1~14; 32 AetINVs were named AetCWI1~15, AetVI1~11, and AetCI1~6 (Appendix A).

In the CI-α and CI-β group, the given genes from monocotyledonous and dicotyledonous plants were clustered on the same branch, confirming that TaCIs were formed before the differentiation of monocotyledonous and dicotyledonous plants. In the VI group, TaVI1, TaVI2, and TaVI3 were very close to OsVIN1 and ZmINVVR1, and the other 41 TaVIs homologous to OsVIN2, indicated that *TaVI1*, *TaVI2*, and *TaVI3* genes evolved earlier than other *TaVI* genes (Figure 1). The CWI-α group contained all dicotyledonous genes and some monocotyledonous genes, and the CWI-β group only contained monocotyledonous genes. This indicates that the *TaCWI* gene existed before the monocotyledonous differentiation and was amplified in a large amount in the monocotyledonous plants after the monocotyledonous differentiation (Figure 1).

### 3.3. Syntenic Analysis of TaINV Genes

In order to further infer the evolutionary origin and homology of the wheat INV family, we constructed collinear chart comparing seven species with wheat, including five monocotyledons (*T. dicoccoides*, *T. urartu*, *Ae. tauschii*, *Z. mays*, and *O. sativa*) and two dicotyledons (*A. thaliana*, and *G. max*) (Figure 2). Wheat and Arabidopsis, soybean, and rice and maize have fewer collinear gene pairs, which are 10, 26, 30 pairs of homologous genes, respectively (Figure 2A and Figure 2B). While more gene pairs were found in wheat and *T. urartu*, *Ae. tauschii*, and *T. dicoccoides* (170, 63, and 78 pairs of homologous genes, respectively) (Figure 2C). Compared with *T. urartu*, and *Ae. tauschii*, judging from the number of orthologous INV gene pairs provided by common wheat and its sub-genome, we found that wheat INV genes are more derived from *T. dicoccoides* (Figure 2D). In addition, among the three sub-genomes (A, B and D) of wheat, 84 orthologous gene pairs were identified, including 26 pairs between A and B sub-genomes, 20 pairs between B and D, 32 pairs between A and D, and 6 pairs between 4A and 7A (Appendix A). The number of pairs of this gene is comparable to that of common wheat and its subgenomic donors, indicating that polyploidization is the main cause of *TaINV* gene expansion in wheat. In addition, 34 *TaINVs* were identified as tandem duplication genes that formed 17 tandem duplication pairs (Figure 3).

The Ka/Ks ratios of *TaINV* duplications genes varied from 0 to 0.8509, indicating that the duplicated *TaINV* gene has undergone purification selection (Appendix A). The divergence time of tandem gene pairs ranged from 13.93 to 53.06 Mya (with an average of 35.62 Mya), while from 0.91 to 56.99 Mya in segmental duplication gene pairs (with an average of 12.11 Mya). This indicates that tandem duplication gradually decreases with the evolution, while the event of fragment duplication between chromosomes is still occurring.

### 3.4. Chromosomal Location Analysis of TaINVs

Except for *TaCWI60* and *TaCWI61*, which are located in scaffolds, the remaining 124 *TaINVs* were distributed on the remaining 20 chromosomes except 7B (Figure 3). In general, the number of *TaINV* genes in A-subgenome is 51 at most, followed by 41 in D-subgenome, and at least 32 in B-subgenome. Most notably, *TaCIs* are evenly distributed in homologous groups 2, 3, 4, and 6, and except for one gene on each of the 3A, 3B, and 3D chromosomes, there are two *TaCIs* located on the remaining chromosomes, respectively. Most *TaVIs* were clustered and located at the ends of chromosomes, except that *TaVI1*, *TaVI2*, and *TaVI3* are located at 2A, 2B, and 2D, respectively. The other 41 *TaVIs* are distributed in five regions with high gene density. Specifically, there are 3, 4, 4, 9, 10, 11 *TaINVs* at 6AS, 6BS, 6DS, 4AL, 7AS, and 7DS, respectively (Figure 3). All *TaCWIs* were distributed on 18 chromosomes except the seventh homologous group. The homologous group 2 has the largest distribution of 20 genes. The homologous group 1 has the least distribution—only three genes. The uneven distribution may be attributable to differences in the size and structure of the chromosomes.

### 3.5. Gene Structure and Cis-Acting Elements Analysis of TaINVs 

Analyses of gene structures showed that the exon/intron distribution patterns were similar among INV members within each clade of the phylogenetic tree, but there are some differences in the lengths of the introns among genes (Figure 4C). The exon–intron pattern of the *TaCI* genes is relatively conservative. Genes in the CI-α group contain 6–8 exons, while in the CI-β group they contain 4–5 exons. It is worth noting that the genes in CI-α group contain a smaller exon encoding 16 amino acids. Unlike the *TaCI* genes, acid invertases have more types of exon–intron structure. Specifically, *TaVIs* have 2–7 exons and the *TaCWIs* have 3–11 exons. The 4 and 7 exons structure genes account for the majority of *TaVIs* and *TaCWIs*, respectively. It is noted that the first exon was longer in *TaVI* genes than *TaCWI* genes (Figure 4C).

To understand the regulative elements of INVs, the cis-acting elements of the promoter region among 126 *TaINVs* were analyzed. The results showed that the cis-acting elements present in the promoter region of the *TaINV* genes could be divided into 10 categories: light-responsive, abscisic acid responsiveness, anaerobic induction, meristem expression, gibberellin-responsiveness, drought-inducibility, anoxic specific inducibility, endosperm expression, differentiation of the palisade mesophyll cells, root-specific expression, and other elements. Among them, the cis-acting elements related to light-responsive were particularly abundant, and there were 124 genes with light-responsive elements, except for *TaCWI3* and *TaCWI53*. The hormone response elements are the most common, with 117 and 68 genes related to ABA and GA, respectively. In addition, 21 genes contain an endosperm-specific expression of cis-acting elements, 70 genes contain CCAAT-box, which is the MYBHv1 binding site, while only *TaCWI31* has root-specific cis-acting elements (Figure 4B).

### 3.6. Features Prediction and Conserved Motifs Analysis of TaINV Proteins

The deduced invertase proteins of the 126 TaINV ranged from 456 to 756 amino acid residues, and their MW range from 51.37 to 84.88 kDa, similar to the invertase in its ancestors (Appendix A). In addition, the putative proteins encoded by *TaVI*, *TaCWI,* and *TaCI* hold pI ranged from 4.77 to 6.91, 4.69–9.28, and 5.35–8.24, respectively (Appendix A). The grand average of hydropathicit (GRAVY) of the TaINVs proteins varied from −0.109 to −0.444, indicating that they were all hydrophilic proteins. The protein instability index shows that all TaCIs were unstable proteins with the Instability index from 42.65 to 54.65, while most of TaVIs (36 out of 44) were stable proteins. For TaCWIs, more than half of TaCWIs (36 of 61) were predicted to be stable.

To better understand TaINVs, the MEME tool was used to search for the conserved motifs in acid invertase and heavy invertase. In acid invertase clade, 10 conserved motifs were identified. Motif 1 (YXXØ motif) is a tyrosine-based lysosomal sorting signal and specifically exists at the N terminal of all TaVIs. The existence of P/V in WECP/VD in other species distinguishes CWI and VI, which becomes more complex in wheat. There is P or L in TaCWI and V or I in TaVI in wheat EC motif (motif 6). Motif 7 has four conserved hydrophobic amino acid (G, A, F, and D) and two consecutive conserved hydrophilic amino acid (RR), which may have an important role in the correct folding of proteins. Motif 2, known as β-fructosidase motif, is found lacking in TaCWI10, TaCWI13, TaCWI21, TaVI29, and TaVI39. Motif 6 (EC motif) contained a catalytic site E that lacked in TaCWI58. Thus, we proposed that the enzyme catalytic activity encoded by these six genes is reduced or lost (Figure 5).

In TaCI class, all genes contain the 10 motifs, with little difference in some amino acids. Aside from TaCI1 that lacks motif 10 and TaCI7 that lacks motif 1, the remaining 19 TaCIs contain all 10 motifs. Notably, six conserved amino acid residues (Leucine/Phenylalanine, L/F in motif1; Glutamic acid/Aspartic acid, E/D and Threonine/Methionine, T/M in motif4; Valine/Isoleucine, V/I in motif6; Tyrosine/Phenylalanine, W/F in motif7; and Proline/Arginine, P/R in motif9) from five conserved motifs were regular varied within the CI gene family (Appendix A). These six pairs of amino acid substitutions could divide TaCIs into CI-α clade and CI-β clade (Figure 6).

### 3.7. The Expression Pattern Analysis of TaINV Genes

Gene expression patterns often contain clues to the function of the gene. The transcriptome results showed that the *TaINV* genes exhibited different temporal–spatial expression patterns (Figure 7) and had different response patterns to stress (Appendix A). There was a total of 70 *TaINV* genes involved in stress response, including 14 *TaCI* genes, 26 *TaCWI* genes, and 30 *TaVI* genes. Among them, 35 *TaINV* genes specifically respond to one of the 6 stresses (drought, heat, cold, *Pst*, *Bgt,* and *Fhb*). For example, *TaVI20*, *TaCWI33*, *TaCWI12*, *TaCI21*, *TaCI14*, and *TaCWI41* were specifically upregulated for drought, heat, cold, *Pst*, *Bgt*, and *Fhb* stress, respectively (Appendix A). *TaCI11*, *TaCWI16*, *TaCWI14*, and *TaVI33* were downregulated for heat, cold, *Bgt*, and *Fhb* stress, respectively. The remaining 35 genes respond to more than two kinds of stress. Intriguingly, *TaVI30* could be regulated by drought, cold, heat, *Pst,* and *Bgt* stress, but the expression patterns showed different upregulated levels. Additionally, *TaVI35* was upregulated in response to drought, cold, heat, and stress, but downregulated in response to *Bgt. TaCWI40* was upregulated by *Pst* and *Fhb*, and downregulated by cold stress. These differential genes provide a cue for the research of plant resistance to biotic and abiotic stresses.

As for temporal-spatial expression patterns, 22 genes were expressed in all tissues and developmental stages, and 20 genes were not expressed in all tissues and developmental stages. The remaining 84 genes exhibited variable expression patterns. Among them, 19 *TaINVs* showed tissue-specific expression. In detail, *TaCI4* and *TaCI20* were specifically expressed in leaf and spike, respectively. *TaCWI17*, *TaCWI18*, *TaCWI24*, *TaCWI53*, and *TaCWI59* were specifically expressed in roots. 12 genes (*TaCWI1*, *TaCWI2*, *TaCWI3*, *TaCWI10*, *TaCWI19*, *TaCWI30*, *TaCWI50*, *TaVI9*, *TaVI10*, *TaVI26*, *TaVI27*, and *TaVI36*) exhibited grain-specific expression (Figure 7).

### 3.8. The Difference in the Expression of TaINV Genes between Grain Weight NILs

Since a large number of genes (12 of 19) are specifically expressed in grains, we speculate that *TaINVs* may play an important role in grain development. Considering the limitation of the tested grain development timepoints in previous literature and the existing relationship between grain with other tissues, we selected nine different expressed genes to test the expression pattern in grain size NILs. *TaVI13* and *TaCI7* were not differentially expressed between RHL81-L and RHL81-S. *TaCI7*, *TaVI27*, *TaCWI47*, and *TaCWI50* were significantly highly expressed in RHL81-L, while *TaVI11*, *TaCWI48*, and *TaCI8* were significantly highly expressed in RHL81-S at 4 DPA. *TaVI11* and *TaCWI47* were significantly highly expressed in RHL81-S at 10 DPA. It is worth noting that the expression of *TaCWI50* and *TaVI27* in RHL81-L was more than ten-fold that of RHL81-S. This may indicate that *TaCWI50* and *TaVI27* play a more important role in grain development (Figure 8).

### 3.9. Subcellular Location Analysis of Wheat INV Proteins

According to the prediction results of ProtComp website, the protein localization in each category is consistent and different from the other two categories, that is, all TaCWI is located outside the cell, all TaVI is located in the vacuole, and all TaCI is located in the cytoplasm. In detail, most of the TaCWIs (34 out of 61 proteins) are typical secretory proteins due to containing a typical signal peptide. However, some differences between CWI protein sequences of the invertase subfamily are obviously. Since *TaCWI50*, a signal peptide-absented INV, was expressed more highly in large grain lines than in small grain lines, we studied its subcellular localization. The results showed that TaCWI50 was located on the cell membrane. This is inconsistent with the software prediction results (Figure 9) and indicated that the members of INVs have multiple functions in wheat.

## 4. Discussion

The INV genes were widely distributed in plant species, such as Arabidopsis [5], rice [6], maize [7], and soybean [8]. The INV play important roles in multiple processes of plant growth and resistance against environmental stressors [51]. In this study, we found that the *TaINVs* distributions among different chromosomes are imbalanced. The chromosome 4A with 16 *TaINV* genes has the highest number of any chromosome. Aside for *TaCWI10,* which is located in 4AS, the other 15 genes are located on chromosome 4AL. Among them, *TaCWI32* collinears with *TaCWI37* and *TaCWI39*, showing the collinear relationship between 4AL/4BS/4DS. Four *TaCWIs* (*TaCWI33* to *TaCWI35*) were collinear with genes located in 5BL and 5D. As for *TaVIs*, except for *TaVI6*, *TaVI10*, and *TaVI12*, the remaining six genes collinear with genes in 7AS and 7DS. The collinearity between chromosomes of different homologous group may be due to the reciprocal translocations during wheat information [52].

Intriguingly, a total of 120 pairs of gene pairs was identified between 106 *TaINVs* and 23 *TuINVs*, 67 *TdINVs,* and 30 *AetINVs*. The 106 *TaINVs* includ all *TaCIs*, 36 *TaVIs*, and 49 *TaCWIs.* We speculated that the other 20 *TaINVs* were generated by fragmented replication and tandem repeat replication according to the intraspecies collinearity analysis of wheat. In detail, 10 *TaINVs* including *TaCWI7*, *TaCWI12*, *TaCWI17*, *TaCWI52*, *TaCWI58*, *TaVI12*, *TaVI21*, *TaVI23*, *TaVI25*, and *TaVI29* were generated by the tandem repeat of *TaCWI6*, *TaCWI13*, *TaCWI16*, *TaCWI51*, *TaCWI57*, *TaVI11*, *TaVI20*, *TaVI22*, *TaVI24*, and *TaVI28,* respectively. *TaCWI10*, *TaCWI44*, and *TaVI29* were generated by the fragmented replication of *TaCWI19*, *TaCWI33,* and *TaVI7* genes, respectively (Figure 2, Appendix A). Taken together, the polyploidy event of wheat species formation may be the main reason for the amplification of the *TaINV* gene family. Furthermore, the tandem and segmental duplication events contributed to the expansion of the acid invertase gene family in wheat.

The known functions of gene family members can be used to predict the functions of other genes on the same branch. Collinearity and phylogenetic tree analysis showed that *TaVI14*, *TaVI16*, and *TaVI21* are orthologous to the *OsVIN2* gene. Tissue-specific expression analysis showed that the expression levels of these three genes were highest in the flag-leaf panicle, which was similar to the expression pattern of the *OsVIN2* gene. Therefore, we speculate that these three genes may also play an important role in wheat yield formation. Similarly, *OsCIN1* is associated with early grain filling regulating in rice [53], while *OsCIN2* regulates grain shape and weight [26]. Here, we noted that *TaCWI4* and *TaCWI22*, the orthologous genes of the *OsCIN1* gene, have the highest expression level in grain at the 2 DPA, which have the same expression pattern with *OsCIN1* (specifically expressed 1–4 days after flowering). Thus, we deduced that *TaCWI4* and *TaCWI22* are important for supplying a carbon source to developing filial tissues by cleaving unloaded sucrose in the apoplast. Further, *TaCWI14*, the orthologous genes of *OsCIN2* gene, may also be a potential domestication gene and that such a domestication-selected gene can be used for further crop improvement. Here, the tissue expression profile analyses of the *TaINV* genes showed that many *TaINV* genes were highly expressed in spike and grain, suggesting that these genes may play important roles in regulating early grain filling and grain size. Although the detailed functions need to be tested in wheat, this provided a cue for dissecting the grain yield.

The expression levels of two genes in tandem duplication gene pairs usually exhibited expression discrepancy in animal and model plants, indicating that the retention of gene duplicates might be associated to processes of tissue expression divergence [54,55]. In this study, four tandem gene pairs showed the same expression pattern. In details, *TaVI7*/*TaVI8* and *TaVI28*/*TaVI29* gene pairs are not expressed in all tissues, while *TaVI9/TaVI10* and *TaVI26/TaVI27* are specific expressed in grains. Additionally, the four gene pairs do not respond to any stress. The other gene pairs had different expression patterns. Four genes including *TaCWI13*, *TaCWI26*, *TaCWI52*, and *TaCWI58* show no functionalization in four gene pairs (*TaCWI12/13*, *TaCWI25/26*, *TaCWI51/52*, and *TaCWI57/58*). 

However, the expression levels of 13 pairs of tandem duplication genes exhibited expression discrepancy. For instance, *TaVI21* was highly expressed in 15 different tissues, but the expression level of *TaVI20* was low. Similarly, *TaVI21* was strongly downregulated under cold stress, and *TaVI20* did not response to cold stress. For plant development, *TaCWI47* and *TaCWI48* were differentially expressed with opposite expression patterns at DPA4 in RHL81L/S NILs, while *TaCWI49* had anther-specific expression [38] and *TaCWI50* had grain-specific expression. This information gave us a cue that tandem duplication of *TaINVs* may be not only function in increasing levels of tissue specificity but also in genomic complexity, and therefore in increasing the adaptation of plants.

In summary, we proposed that *TaCWI50* and *TaVI27* could be candidate genes relating to the grain-size trait. Although much work needs to be done to verify the detailed function in the future, this study here set an important base to understanding *TaINV* genes in the regulation of growth, development, and tolerance of various stresses.

## 5. Conclusions

In this study, we comprehensively characterized 126 *TaINV* genes and categorized them into five classes. The number of INV genes has increased dramatically in wheat, and we speculate that polyploidy events are the main force behind its expansion. Compared with CI, TaCWI and TaVI are more similar in gene structure and protein properties. *TaINVs* have different spatiotemporal expression patterns and play an important role in responding to biotic and abiotic stresses. Our results demonstrated that TaCWI50 was located on the cell membrane and its high expression may be associated with a larger TGW phenotype.

## Figures and Tables

**Figure 1 genes-14-00041-f001:**
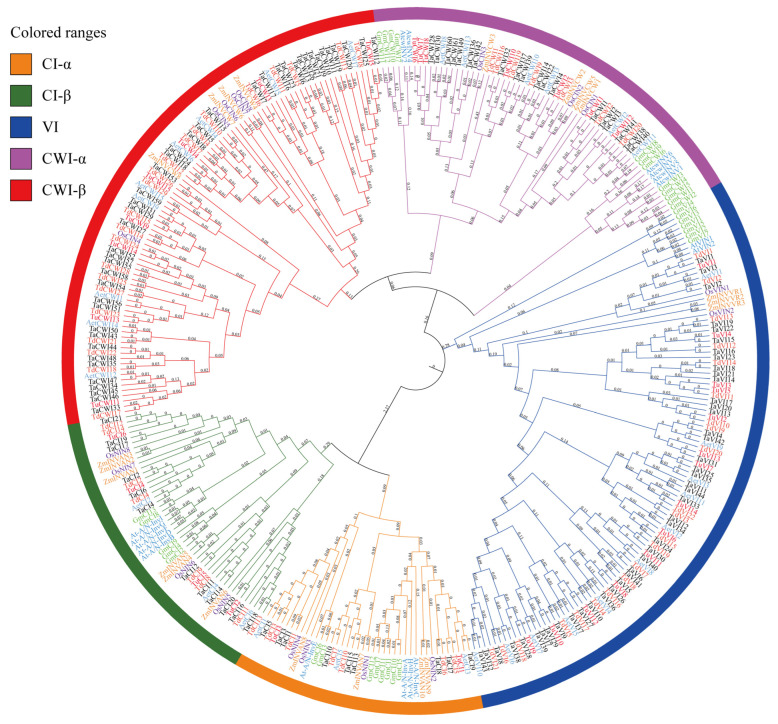
Phylogenetic analysis of the invertase family members in *A. thaliana*, *G. max*, *O. sativa*, *Z. may*, *T. urartu*, *Ae. tauschii*, *T. dicoccoides*, and *T. aestivum*. The INVs from different species are labeled with different colors.

**Figure 2 genes-14-00041-f002:**
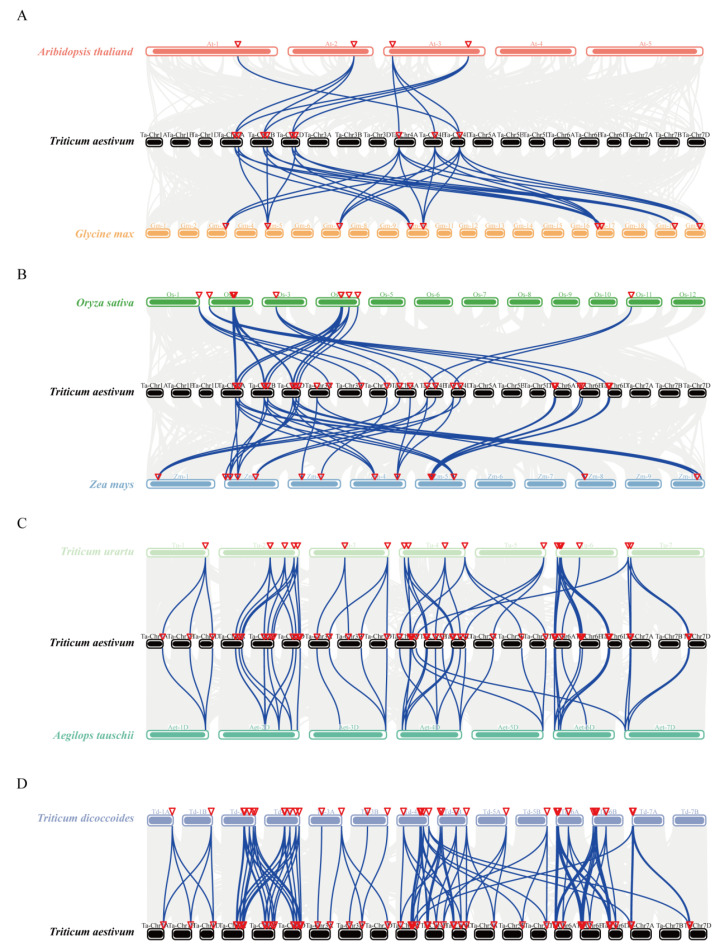
Synteny analysis of INV genes between wheat and two dicotyledons plants (*A. thaliana*, and *G. max* (**A**), two monocotyledons model plants (*Z. mays*, and *O. sativa*) (**B**), its relative ancestors (*T. urartu*, *Ae. tauschii* (**C**), and *T. dicoccoides*(**D**)). The gray line in the background indicates a collinear block, while the blue line highlights the isomorphic INV gene pair.

**Figure 3 genes-14-00041-f003:**
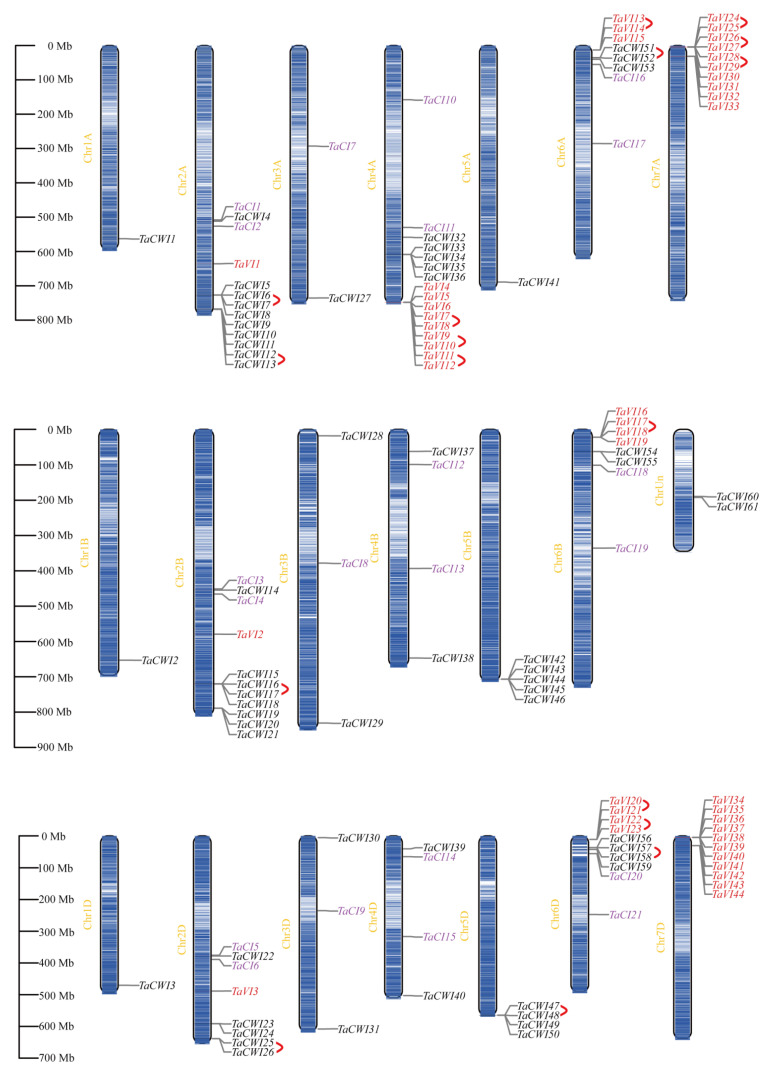
Chromosomal localization of *TaINVs*. Different groups of *TaINVs* are represented in different colors. Black represents CWI group, red represents VI group, and blue represents CI group. In addition, tandem repeat genes are connected with red brackets.

**Figure 4 genes-14-00041-f004:**
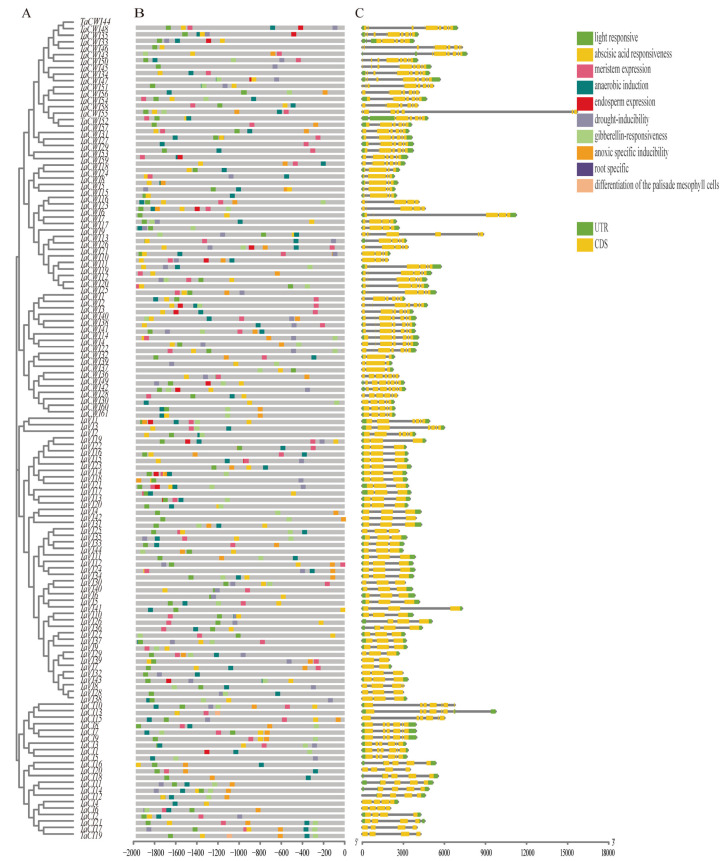
Phylogenetic relationship(**A**), cis-acting elements (**B**) and gene structure analysis (**C**) of TaINVs.

**Figure 5 genes-14-00041-f005:**
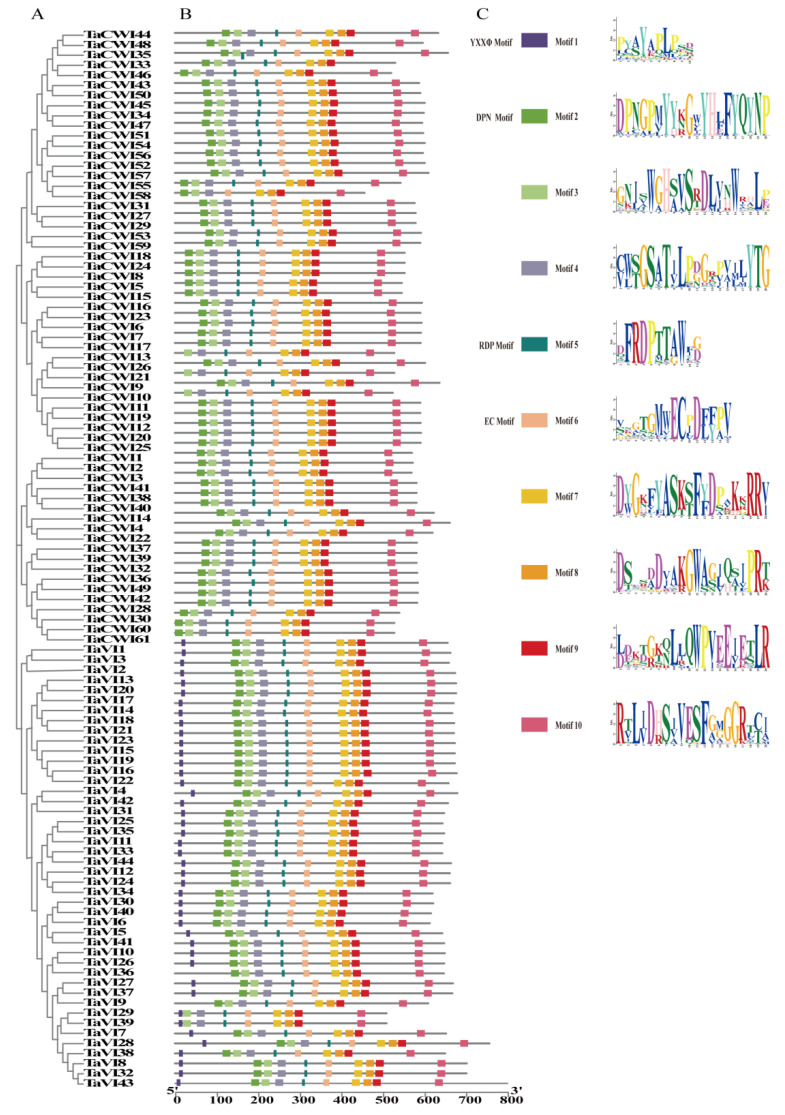
The motifs distribution of wheat Ac-INV proteins (**B**) were displayed according to phylogenetic relationships (**A**). The logos of ten motif were shown proportionally (**C**).

**Figure 6 genes-14-00041-f006:**
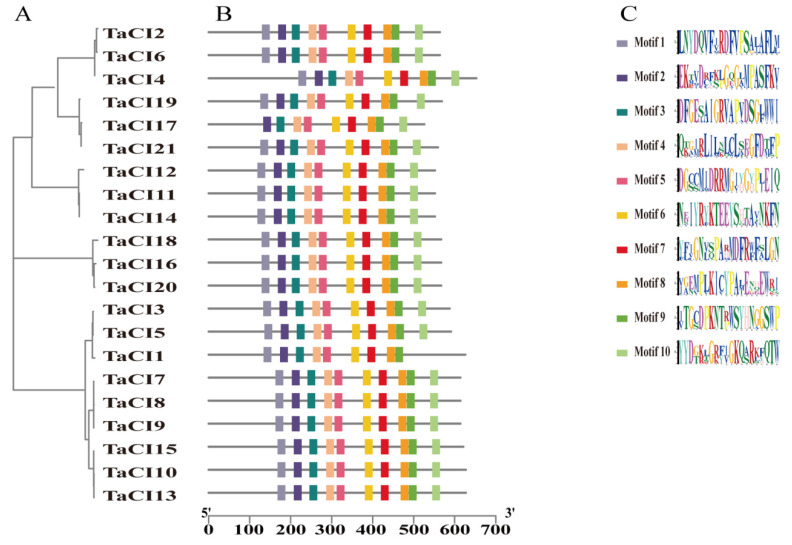
The motifs distribution of TaCI proteins (**B**) were displayed according to phylogenetic relationships (**A**). The logos of 10 motif were shown proportionally (**C**).

**Figure 7 genes-14-00041-f007:**
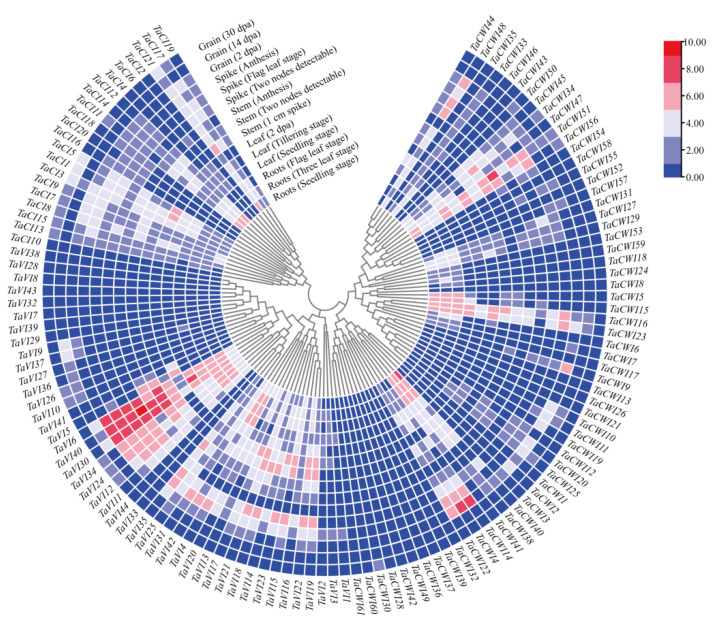
The expression profiles of *TaINV* genes involved in 5 tissues at different growth stages. The color scale of heatmap shows the level of gene expression.

**Figure 8 genes-14-00041-f008:**
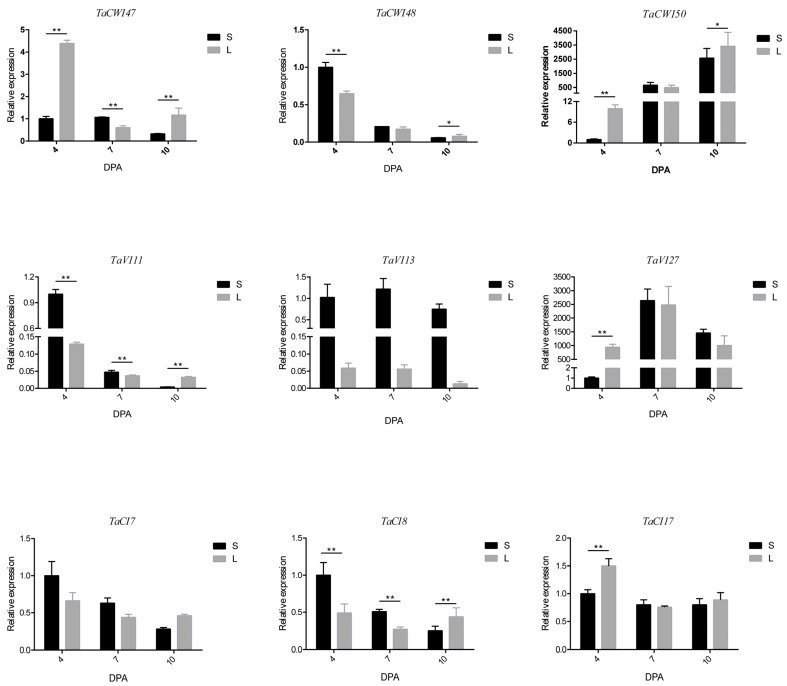
Expression profiling of 9 *TaINV* genes in grains at D4, D7, and D10 stage between RHL81-L (L) and RHL81-S (S). Asterisks indicate significant differences between L and S at different time using two-tailed Student’s *t*-test: *p* < 0.01 (**), *p* < 0.05 (*).

**Figure 9 genes-14-00041-f009:**
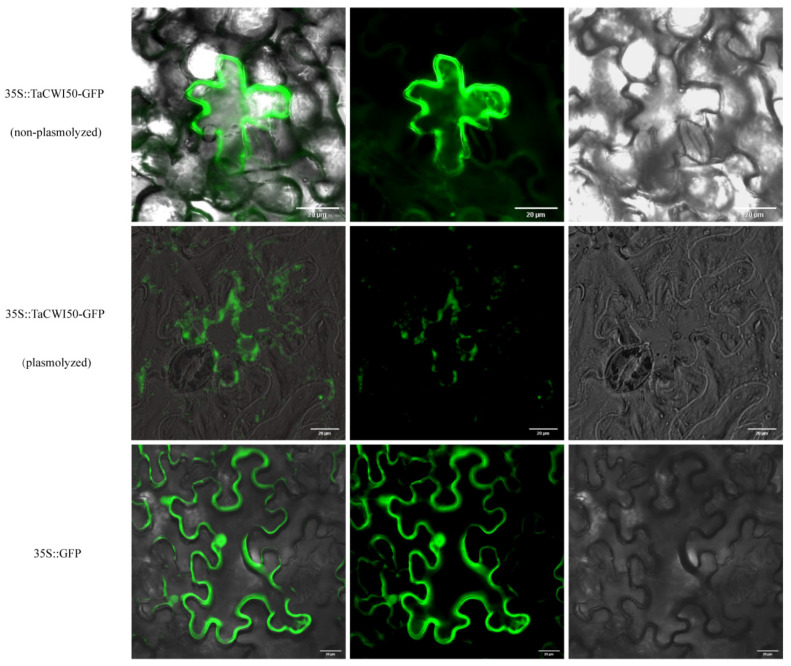
Localization of fusion protein (pCaMV35S::TaCWI50-GFP). Scale bars are shown in the lower right corner of each image.

## Data Availability

Not applicable.

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
