# Peer review of "Characteristics and Expression Analysis of Invertase Gene Family in Common Wheat (Triticum aestivum L.)"

_genes, 2022, doi:10.3390/genes14010041_

Round 1

Reviewer 1 Report

The MS reports insilico characterization of 126 TaNVs. Although 9 genes out of 126 TaINV genes are partly validated but still the work has some significance. Please improve the revised version addressing the following and some other related shortcomings

-Line 45: Please change "The invertase genes were widely existed and played various roles in plants" to "The invertase genes perform various roles in plants"

-Line 74, Please italicize 'T. aestivum

-Line 86-88: Regarding "Meanwhile, the protein sequences of 86 17 INVs in Arabidopsis and 18 INVs in rice were used as queries to search against the 87 local protein database using an e-value cutoff of 1e-10 and an identity of 75% as the thresh- 88 old. "' are you sure of the cut off values. What was the querry coverage?

Line 103: Please cite this software Dual Systeny Plotter 

Author Response

Point 1: Line 45: Please change "The invertase genes were widely existed and played various roles in plants" to "The invertase genes perform various roles in plants"

Response 1: Modified as required (line 51 in the current version).

Point 2: Line 74, Please italicize 'T. aestivum”

Response 2: Modified as required.

Point 3: Line 86-88: Regarding "Meanwhile, the protein sequences of 86 17 INVs in Arabidopsis and 18 INVs in rice were used as queries to search against the 87 local protein database using an e-value cutoff of 1e-10 and an identity of 75% as the thresh- 88 old." are you sure of the cut off values. What was the querry coverage?

Response 3: Modified as required. The identity value in line 88 (line 115 in the current version) mentioned in the question is incorrectly written, and "75%" has been changed to "50%". The querry coverage > 70% was used as threshold and which is supplemented in line 114 of the manuscript.

Point 4: Line 103: Please cite this software Dual Systeny Plotter

Response 4: Modified as required. Since the Dual Systeny Plotter software is a plug-in in TBtools, it is cited at the first occurrence of line 103, and the subsequent references to TBtools in line 131 are deleted.

Reviewer 2 Report

Title: Characteristics and Expression Analysis of Invertase Gene 2 Family in Common Wheat (Triticum aestivum L.)

The authors have described the Invertase (INV) gene family in wheat which irreversibly catalyzes the conversion of sucrose into glucose and fructose, playing important role in plant development and stress tolerance. The information provided in this manuscript will help the scientific community to understand wheat's grain development mechanism. The study is well-described, and the following are the general comment to be considered while revising this manuscript.

1. Please provide the expanded form of abbreviated words and acronyms when using them for the first time.

2. Introduction section can be more elaborate to focus on the questions beings addressed in this study.

3. This manuscript requires minor revision.

Author Response

Point 1: Please provide the expanded form of abbreviated words and acronyms when using them for the first time.

Response 1: Modified as required. 

“CS ” in line 89 has been changed to “Chinese Spring (CS)”.

“Ka/Ks ” in line 136 has been changed to “nonsynonymous (Ka)/synonymous (Ks)”.

“CDS ” in line 191 has been changed to “coding sequence (CDS)”.

Point 2: Introduction section can be more elaborate to focus on the questions beings addressed in this study.

Response 2: The following supplements have been made.

Add “The invertase genes have been identified in many plant species, including Arabidopsis thaliana [5], Oryza sativa [6], Zea may [7], Glycine max [8], Saccharum officinarum [9], Pisum sativum [10], Solanum tuberosum [11], Malus domestica [12], and Populus [13].” in line 39.

Add “Ac-Invs contains three conserved motifs of NDPNG (β-fructofuranosidases motif), RDP, and WECP(V)D, which are necessary for the catalytic activity [6]. Moreover, three amino acids (DPN) of the β-fructofuranosidases motif were coded by the smallest exon known to plant species [14].” in line 44.

Add “The CIs can be further divided into CI-α and CI-β subgroups.” in line 50.

Point 3: This manuscript requires minor revision.

Response 3: The following minor modifications have been made.

1. The institute of first author and co-corresponding authors has been adjusted in line 5.

2. Change the “importance”to ”complexity”in line 92.

3. Line 112 adds the threshold of hmmsearch, which is "with a threshold of e < 1e−10".

4. As the indicator line of TaVI34gene on 7D chromosome in Figure 3 is misplaced, the Figure 3 is modified and replaced.

5. The references in the manuscript have been revised due to the new references cited in the manuscript.

Reviewer 3 Report

This article studied Characteristics and Expression Analysis of Invertase Gene Family in Common Wheat (Triticum aestivum L.). There are some shortcomings for that should be resolve.

Line 18 different spatiotemporal expressions should be specify.

Highlight methods or techniques in the abstract.

Line 45 specify its role in different plants.

Discuss commercial importance of wheat and how the analysis in wheat will be helpful for genetics

Line 74 italicize the species name.

I suggest that Section 2.2 should be cited with relevant studies.

https://doi.org/10.1016/j.plaphy.2021.01.042, https://doi.org/10.1007/s10725-021-00785-7,

Conclusion is well justified. The authors should discuss some points for the future studies molecular level studies are required to know about the involved mechanism and improvement of invertase breeding. 

Author Response

Point 1: Line 18 different spatiotemporal expressions should be specify.

Response 1: Modified as required.

Change “TaINVs have different spatiotemporal expression patterns and play important roles in responding to biotic and abiotic stresses. Among them, nine TaINVs were selected and validated by qRT-PCR in grain weight NILs. Our results demonstrated that TaCWI50 was located on the cell membrane and its high expression may be associated with a larger TGW phenotype.” to “TaINVs expressed in multiple tissues with different expression levels. Among 19 tissue-specific expressed TaINVs, 12 TaINVs show grain-specific expression pattern and may play an important role in wheat grain development. In addition, qRT-PCR results further confirmed that TaCWI50 and TaVI27 show different expression in grain weight NILs. Our results demonstrated that the high expression of TaCWI50 and TaVI27 may be associated with a larger TGW phenotype.”

Point 2: Highlight methods or techniques in the abstract.

Response 2: Modified as required.

Add “using a genome-wide search method” in line 13.

Add “in the synteny analysis” in line 15.

Point 3: Line 45 specify its role in different plants.

Response 3: Modified as required.

Add “Mutation of CYT-INV1 in Arabidopsis and rice resulted in a shorter primary root [19, 21]. Similar to CIs, the role of VI in root development has also been found. In detail, allelic differences in AtVI2 cause differences in root length between divergent genotype [22]. As for grain development and yield formation, mutation of the CWI gene INCW2 in maize reduced maize grain weight by 70% [23], and mutation of OsINV3 lead to a shorter panicle with lighter and smaller grains [24]. Conversely, over expression of CWI enhanced grain filling and yield in rice and maize [25, 26]. INVs are also play important roles in abiotic responses. CsVI2 was up-regulated by drought and the overexpression of CsVI2 could enhanced drought tolerance in cucumber seedlings [27] ” in line 53.

Delete “CWIs, involved in phloem unloading, played an important role in defence response and yield formation [5, 16-18]. VIs had roles in the determination of hexose-to-sucrose ratio [19], sink strength [20], and seed size [21]. ” in line 52. Meanwhile, add “AtVI2 may play an important role in regulating stomatal opening in Arabidopsis [30]. Knockdown of vacuolar invertase gene expression lowers the cold-induced sweetening in potatoes [31].” in line 52.

Point 4: Discuss commercial importance of wheat and how the analysis in wheat will be helpful for genetics.

Response 4: Modified as required. Add “As the third largest cereal crop, common wheat accounting for approximately 20 % of the total dietary calories and proteins worldwide [33]. With the continuous growth of the population, the demand for wheat is increasing, so improving the yield has always been one of the main tasks of wheat breeding [34]. The new yield related gene mined is helpful to improve wheat yield through gene editing technology [35]. While, genome-wide gene family analysis laid the foundation to excavate key genes [36]. " in line 75.

Add “This will lay a foundation for the functional analysis of TaINV gene in the future.” in line 96.

Point 5: Line 74 italicize the species name.

Response 5: Modified as required.

Point 6: I suggest that Section 2.2 should be cited with relevant studies. 

https://doi.org/10.1016/j.plaphy.2021.01.042, https://doi.org/10.1007/s10725-021-00785-7,

Response 6: Thank you for your suggestions. These two studies are relevant to my research and have been cited in line 123.
